# LOWER BOUNDS ON THE ROBUSTNESS OF FIXED FEATURE EXTRACTORS TO TEST-TIME ADVERSARIES

## ABSTRACT

Understanding the robustness of machine learning models to adversarial examples generated by test-time adversaries is a problem of great interest. Recent theoretical work has derived lower bounds on how robust *any model* can be, when a data distribution and attacker constraints are specified. However, these bounds only apply to arbitrary classification functions and do not account for specific architectures and models used in practice, such as neural networks. In this paper, we develop a methodology to analyze the robustness of fixed feature extractors, which in turn provides bounds on the robustness of any classifier trained on top of it. The tightness of these bounds relies on the effectiveness of the method used to find collisions between pairs of perturbed examples at deeper layers. For linear feature extractors, we provide closed-form expressions for collision finding while for arbitrary feature extractors, we propose a bespoke algorithm based on the iterative solution of a convex program that provably finds collisions. We utilize our bounds to identify the layers of robustly trained models that contribute the most to a lack of robustness, as well as compare the same layer across different training methods to provide a quantitative comparison of their relative robustness.

## 1 INTRODUCTION

The robustness of machine learning models to test-time (evasion) attacks, particularly via adversarial examples, has been studied extensively from an empirical perspective (Croce et al., 2020; Papernot et al., 2016; Li et al., 2020; Biggio & Roli, 2017). A plethora of attacks (Szegedy et al., 2013; Carlini & Wagner, 2017; Bhagoji et al., 2018; Croce & Hein, 2020) and defenses (Madry et al., 2018; Zhang et al., 2019) has been proposed. The resulting arms race between proposed attacks and defenses, has led recent theoretical work (Dohmatob, 2019; Cullina et al., 2018; Mahloujifar et al., 2019; Bhagoji et al., 2019; 2021; Montasser et al., 2019) to focus on understanding the fundamental bounds of learning in the presence of adversarially perturbed inputs, in terms of both the existence of robust classifiers and the sample complexity required to learn them.

A line of research on this front has investigated lower bounds on the robustness attainable by *any classifier* when classifying data from distributions modified by adversarial perturbations (Pydi & Jog, 2020; Bhagoji et al., 2019; 2021). In particular, information-theoretic lower bounds on the minimum possible robust $0 - 1$ and cross-entropy losses have been derived for arbitrary discrete distributions and very general attack models. However, the bounds from these papers are only applicable when considering the set of possible classifiers to be *all measurable functions*, which does not translate to practice. The classifier architecture and even some portion of the parameters are fixed (such as in transfer learning) in practical approaches to training machine learning models.

In this paper, we leverage this line of work to find lower bounds on the robustness of commonly used, fixed feature extractors. This provides a principled method to compare the robustness of feature extractors obtained by different training methods to arbitrary test-time adversaries. The key question we answer is:

*What is the minimum robust loss incurred by any classifier trained on top of a fixed feature extractor?*

Two challenges for training robust machine learning models motivate this question. First, since transfer learning is now common practice, it is incumbent upon the model trainer to choose a robust pre-trained feature extractor. Our work offers a principled and quantitative method to choose among

feature extractors obtained from different layers and training methods. Second, we are able to shed light on the impact of common architectural choices, such as activation functions and dimensionality reduction on robustness. This arises from our study of how the lower bound evolves from layer to layer within deep neural networks.

**Importance of lower bounds on robustness:** To determine classifier-agnostic lower bounds on robustness, earlier work focused on the interaction between points from different classes in the input space when perturbed through the construction of a *conflict graph*. Minimizing an appropriately defined loss function over this graph determines an information-theoretic lower bound on the loss. Intuitively, the denser the graph is, the higher the lower bound. These classifier-agnostic bounds are able to determine the plausibility of robust classification for a given adversary and dataset, and highlight practically relevant perturbation budget.

## 1.1 Contributions

**Lower bounds on robustness for fixed feature extractors:** We significantly extend prior work by deriving lower bounds that depend on the architecture and weights of a fixed feature extractors. We construct a distance function over the input space that depends on the feature extractor in use. Our results have implications for both *analyzing the robustness of already trained classifiers* as well as *determining feature extractors to use for applications such as transfer learning*.

**Bespoke algorithms for collision finding:** For *linear feature extractors* such as the first layer of a neural network, we determine exact, closed form expressions to find collisions. Collision finding for *non-linear feature extractors* is a non-convex optimization problem, for which we propose a custom algorithm. We approach the collision-finding problem with a descent algorithm that solves a sequence of convex optimization problems over polytopes. This algorithm has the structural advantage of maintaining feasibility (for some adversarial budget constraint) at each step. Despite the fact that it does not find a global minimum, the solutions found are useful for obtaining not just approximations but actual lower bounds on optimal adversarial classification loss.

**Empirical findings:** We utilize our method to find numerical lower bounds on the robustness of fixed feature extractors trained on two different datasets with both fully-connected and convolutional architectures. Our results indicate that the use of dimensionality-reducing linear layers as well as the effective compression induced by ReLU activations lead to significantly less robust networks. Further, we confirm the oft-cited observation that mismatches between the adversary's budget at train and test time impacts robustness negatively. Taken together, these findings point towards future design considerations for robust models.

## 2 Deriving lower bounds for fixed feature extractors

In this section, we develop a method for evaluating the robustness of a feature extractor for classification in the presence of a test-time adversary. We characterize the optimal adversarial loss achievable by any classifier that uses the fixed feature extractor as its initial layer. This characterization is based on a conflict graph derived from a discrete data distribution and the constraints of the adversary.

**Examples and labels:** We consider a supervised classification problem with a test-time adversary. We have an example space $\mathcal{X}$ and a label space $\mathcal{Y} = \{-1, 1\}$. Labeled examples are sampled from a joint probability distribution $P$ over $\mathcal{X} \times \mathcal{Y}$.

**Test-time adversary:** The test-time adversary modifies a natural example subject to some constraints to generate an adversarial example that will be classified Goodfellow et al. (2015); Szegedy et al. (2013); Carlini & Wagner (2017). Formally, the adversary samples a labeled example $(x, y)$ from $P$ and selects $\tilde{x} \in N(x)$, where $N : \mathcal{X} \to 2^{\tilde{\mathcal{X}}}$ is the neighborhood function encoding the constraints on the adversary and $\tilde{\mathcal{X}}$ is the space of adversarial examples.[1] For all $x$, $N(x)$ must be nonempty. This definition encompasses the $\ell_p$ family of constraints widely used in previous work.

---

[1] In most (but not all) settings that have previously been studied, $\tilde{\mathcal{X}} = \mathcal{X}$. We believe that making the distinction helps to clarify some of our definitions: their applicability in this general context affects what properties they can be expected to have.

**Measuring adversarial loss:** We consider 'soft' classification functions (or classifiers) that map examples to probability distributions over the classes. These are $h : \mathcal{X} \to \Delta^{\mathcal{Y}}$, where $\Delta^{\mathcal{Y}} = \{p \in \mathbb{R}^{\mathcal{Y}} : p \geq \mathbf{0}, \sum_{y \in \mathcal{Y}} p_y = 1\}$. We measure classification performance with a loss function $\ell : \Delta^{\mathcal{Y}} \times \mathcal{Y} \to \mathbb{R}$, so the expected performance of a classifier $h$ is $\mathbb{E}[\sup_{\tilde{x} \in N(x)} \ell(h(\tilde{x}), y)]$, where $(x, y) \sim P$. In the two class setting, $h(\tilde{x})_{-1} = 1 - h(\tilde{x})_1$, so any loss function that treats the classes symmetrically can be expressed as $\ell(p, y) = \ell'(p_y)$. Additionally, $\ell'$ should be decreasing. Together, these allow the optimization over adversarial examples to be moved inside the loss function, giving $\mathbb{E}\left[\ell\left(\inf_{\tilde{x} \in N(x)} h(\tilde{x})_y, y\right)\right]$. Thus, the adversarial optimization can be analyzed by itself.

**Definition 1.** *For a soft classifier $h$, the correct-classification probability $q_v$ that it achieves on an example $v = (x, y)$ in the presence of an adversary is $q_v = \inf_{\tilde{x} \in N(x)} h(\tilde{x})_y$. The space of achievable correct classification probabilities is*

$$\mathcal{P}_{\mathcal{V},N,\mathcal{H}} = \bigcup_{h \in \mathcal{H}} \left\{ q \in \mathbb{R}^{\mathcal{V}} : \forall (x,y) \in \mathcal{V}, \, 0 \leq q_{(x,y)} \leq \inf_{\tilde{x} \in N(x)} h(\tilde{x})_y \right\}.$$

Bhagoji et al. (2021) characterized $\mathcal{P}_{\mathcal{V},N,\mathcal{H}}$ in the case that $\mathcal{H}$ is the class of measurable functions $\mathcal{X} \to \Delta^{\mathcal{Y}}$. For a data distribution $P$ that is discrete with finite support $\mathcal{V} \subseteq \mathcal{X} \times \mathcal{Y}$, this allows the minimum adversarial loss achievable to be expressed as an optimization over $\mathcal{P}_{\mathcal{V},N,\mathcal{H}}$:

$$\inf_{h \in \mathcal{H}} \mathbb{E}_{(x,y) \sim P} \left[ \sup_{\tilde{x} \in N(x)} \ell(h(\tilde{x}), y) \right] = \inf_{q \in \mathcal{P}_{\mathcal{V},N,\mathcal{H}}} \sum_{v \in \mathcal{V}} P(\{v\}) \ell'(q_v).$$

We extend their approach to analyze feature extractors as follows. Given a feature space $\mathcal{Z}$ and a *fixed, measurable feature extractor* $f : \tilde{\mathcal{X}} \to \mathcal{Z}$, define $\mathcal{H}_f = \{h \in \mathcal{H} : h = g \circ f\}$: an element of $\mathcal{H}_f$ is some measureable $g : \mathcal{Z} \to \Delta^{\mathcal{Y}}$ composed with $f$. Our aim is to characterize $\mathcal{P}_{\mathcal{V},N,\mathcal{H}_f}$ so that we can optimize loss functions over it to evaluate the suitability of $f$.

**Conflict graph:** In Bhagoji et al. (2021), the notion of a conflict graph was used to record neighborhood intersections for pairs of points from different classes. When such an intersection exists, it is impossible for any classifier to correctly classify both of those points in the adversarial setting. We extend this notion to apply to the family of classifiers using a fixed feature extractor $f$. In our setting, a conflict exists between a pair of points when each of them has a neighbor that is mapped to the same point in the feature space.

We call the conflict graph $G_{\mathcal{V},N,\mathcal{H}_f}$, where $\mathcal{V} \subseteq \mathcal{X} \times \mathcal{Y}$. When we apply $G_{\mathcal{V},N,f}$ in order to understand classification of labeled examples with distribution $P$, we take $\mathcal{V}$ to be the support of $P$. The graph is bipartite: the vertex set is partitioned into parts $\mathcal{V}_c = \mathcal{V} \cap (\mathcal{X} \times \{c\})$. The edge set is

$$\mathcal{E}_{\mathcal{V},N,f} = \{((u,1),(v,-1)) \in \mathcal{V}_1 \times \mathcal{V}_{-1} : \exists \tilde{u} \in N(u), \tilde{v} \in N(v) \text{ such that } f(\tilde{u}) = f(\tilde{v})\}.$$

**Lemma 1** (Feasible output probabilities)**.** *The set of correct classification probability vectors for support points $\mathcal{V}$, adversarial constraint $N$, and hypothesis class $\mathcal{H}_f$ is*

$$\mathcal{P}_{\mathcal{V},N,\mathcal{H}_f} = \{q \in \mathbb{R}^{\mathcal{V}} : q \geq \mathbf{0}, \, q \leq \mathbf{1}, \, Bq \leq \mathbf{1}\} \tag{1}$$

*where $B \in \mathbb{R}^{\mathcal{E} \times \mathcal{V}}$ is the edge incidence matrix of the conflict graph $G_{\mathcal{V},N,f}$.*

The proof is given in Appendix A and follows the structure of the proof in Bhagoji et al. (2021).

**Approximating $G_{\mathcal{V},N,f}$ and $\mathcal{P}_{\mathcal{V},N,\mathcal{H}_f}$:** If instead of knowing the true conflict graph $G_{\mathcal{V},N,f}$, we have some subgraph, then we can find a polytope that is a superset of the true $\mathcal{P}_{\mathcal{V},N,\mathcal{H}_f}$. If we minimize some expected loss over this proxy polytope, we obtain a lower bound on the optimal loss over $H_f$. Because subgraphs of the conflict graph lead to valid lower bounds on optimal classification performance, we can use this method to evaluate the quality of a feature extractor $f$ even if exact computation of the conflict graph is computationally intractable.

## 2.1 DISTANCE INTERPRETATION

The construction of a conflict graph and characterization of the feasible correct classification probabilities from the previous section apply to any feature extractor and neighborhood constraint. In

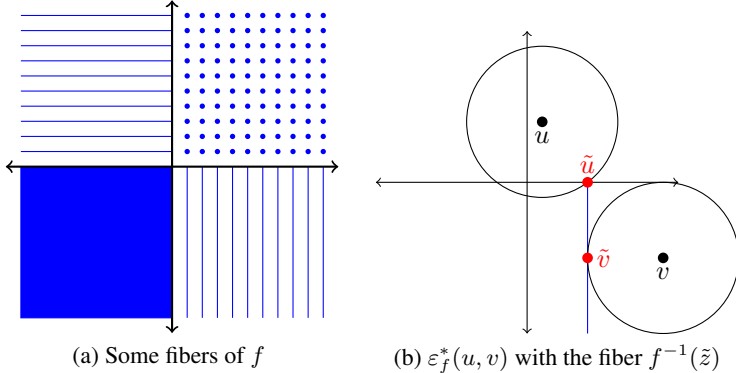

(a) Some fibers of $f$  (b) $\varepsilon_f^*(u,v)$ with the fiber $f^{-1}(\tilde{z})$

Figure 1: Induced distances for $f : \mathbb{R}^2 \to \mathbb{R}^2$, $f(x_0, x_1) = (\max(x_0, 0), \max(x_1, 0))$, (a pair of ReLUs). Let $a, b > 0$. Then the fiber $f^{-1}(\{(a, b)\})$ is the point $\{(a, b)\}$. The fiber $f^{-1}(\{(a, 0)\})$ is the ray $\{(a, y) : y \leq 0\}$. The fiber $f^{-1}(\{(0, 0)\})$ is the quadrant $\{(x, y) : x \leq 0, y \leq 0\}$. For $u = (1, 4)$ and $v = (9, -5)$, we have $\varepsilon_f^*(u, v) = 5$, $\tilde{u} = (4, 0)$, $\tilde{v} = (4, -5)$, and $\tilde{z} = (4, 0)$.

the most commonly studied settings, the neighborhood constraint arises from a distance function: $N_\varepsilon(x) = \{\tilde{x} \in \tilde{\mathcal{X}} : d(x, \tilde{x}) \leq \varepsilon\}$. This parameter $\varepsilon$ is the adversarial budget constraint.

For any two examples $u$ and $v$, a natural quantity to consider is the smallest adversarial budget that would cause the edge $(u, v)$ be appear in the conflict graph:

$$\varepsilon^*(u, v) = \inf\{\varepsilon \geq 0 : \mathbb{N}_\varepsilon(u) \cap \mathbb{N}_\varepsilon(v) \neq \varnothing\} = \inf_{\tilde{x} \in \tilde{\mathcal{X}}} \max(d(u, \tilde{x}), d(v, \tilde{x})).$$

We will call this the distance on $\mathcal{X}$ induced by $d$. When $\mathcal{X} = \tilde{\mathcal{X}} = \mathbb{R}^d$ and $d(x, x') = \|x - x'\|_p$, the minimal adversarial budget is simply $\frac{1}{2}\|u - v\|_p$. Because the relationship to the distance used in the adversarial budget constraint is so simple, this quantity is not usually discussed independently. This definition generalizes easily to the setting of classification with a particular feature extractor, but the resulting quantity is much more interesting.

**Definition 2.** *The distance induced on $\mathcal{X}$ by $d$ and $f$ is the minimum adversarial budget required to create a conflict between $u$ and $v$ after a feature extractor $f$:*

$$\varepsilon_f^*(u, v) = \inf\{\varepsilon \geq 0 : f(N_\varepsilon(u)) \cap f(N_\varepsilon(v)) \neq \varnothing\} = \inf_{\tilde{u}, \tilde{v} \in \tilde{\mathcal{X}} : f(\tilde{u}) = f(\tilde{v})} \max(d(u, \tilde{u}), d(v, \tilde{v})). \quad (2)$$

This reduces to $\varepsilon^*(u, v)$ when $f$ is the identity function, or more generally any injective function. Observe that any choice of $\tilde{u}$ and $\tilde{v}$ in (2) provide an upper bound on $\varepsilon_f^*(u, v)$, which is useful for finding a subgraph of $G_{\mathcal{V}, N, f}$ and lower bounds on optimal classification performance in $\mathcal{H}_f$.

Figure 1 illustrates the computation of $\varepsilon_f^*$ for a simple $f$ that is related to the ReLU function.

**Induced distance is not a distance on the feature space:** The fact $\varepsilon_f^*$ is defined on $\mathcal{X}$ is essential: it is not possible to interpret it as a distance on $\mathcal{Z}$. For a full explanation of this point, see Appendix B.

## 3    DISTANCE COMPUTATIONS FOR PRACTICAL FEATURE EXTRACTORS

Throughout the remainder of the paper, we will work in a more concrete setting. We assume that our example space is a real vector space and that adversarial examples are from the same space: $\mathcal{X} = \tilde{\mathcal{X}} = \mathbb{R}^{n_1}$. Let $\mathcal{B} \subseteq \mathcal{X}$ be a nonempty, closed, convex, origin-symmetric set. These conditions imply the zero vector in contained in $\mathcal{B}$. We take the neighborhood of any point to be a scaled, shifted version of this ball: $N_\varepsilon(x) : x + \varepsilon\mathcal{B}$. Neighborhood constraints derived from $\ell_p$ norms fit into this class: we encode them by defining $\mathcal{B} = \{\delta \in \mathbb{R}^{n_1} : \|\delta\|_p \leq 1\}$, which results in $N_\varepsilon(x) = \{\tilde{x} \in \mathbb{R}^{n_1} : \|\tilde{x} - x\|_p \leq \varepsilon\}$. We will focus on $\ell_2$-based neighborhoods, but our algorithms could be adapted to any choice of $\mathcal{B}$ over which efficient optimization is possible.

### 3.1 LINEAR FEATURE EXTRACTORS

Suppose that our feature extractor is an affine linear function of the input example: $f(x) = Lx + k$ for some matrix $L \in \mathbb{R}^{n_2 \times n_1}$ and vector $k \in \mathbb{R}^{n_2}$. Then the distance $\varepsilon_f(u, v)$ becomes

$$\inf\{\varepsilon \geq 0 : \{k + L(u + \delta) : \delta \in \varepsilon \mathcal{B}\} \cap \{k + L(u + \delta') : \delta' \in \varepsilon \mathcal{B}\} \neq \varnothing\}.$$

Because $\{(\varepsilon, \delta) \in \mathbb{R}^{1+n_1} : \delta \in \varepsilon \mathcal{B}\}$ is closed convex cone, $\varepsilon_f(u, v)$ can be expressed as the following convex optimization problem:

$$\inf \varepsilon \quad \text{subject to} \quad \delta \in \varepsilon \mathcal{B}, \quad \delta' \in \varepsilon \mathcal{B}, \quad L(\delta - \delta') = L(v - u).$$

Because $\mathcal{B}$ is closed and the linear subspace is always nonempty, the infimum is achieved. Also, it is sufficient to consider $(\delta, \delta')$ satisfying $\delta' = -\delta$: for any feasible $(\varepsilon, \delta, \delta')$, the point $(\varepsilon, (\delta - \delta')/2, (\delta' - \delta)/2)$ is also feasible, has the same value, and satisfies the additional constraint. The feasibility of the symmetrized point uses the origin-symmetry of $\mathcal{B}$. The simplified program is

$$\min \varepsilon \quad \text{subject to} \quad \delta \in \varepsilon \mathcal{B}, \quad L\delta = L(v - u)/2.$$

Thus the optimal adversarial strategy for creating conflict between $u$ and $v$ is intuitive: produce examples with the same features as the midpoint $(u + v)/2$.

If $\mathcal{B}$ is the unit $\ell_2$ ball, there are further simplifications. Consider the singular value decomposition $L = U\Sigma V^T$ where we do not include zero singular values in $\Sigma$. Then the linear map given by $U\Sigma$ is injective and can be canceled from the linear constraint on $\delta$. The resulting program is

$$\min \varepsilon \text{ subject to } \|\delta\|_2 \leq \varepsilon, \quad V^T \delta = \frac{1}{2} V^T(v - u),$$

the optimal choice of $\delta$ is $\delta = \frac{1}{2} V V^T(v - u)$, and $\varepsilon_f(u, v) = \frac{1}{2} \|V^T(v - u)\|_2$. Observe that this is the norm of a vector in $\mathbb{R}^{n_2}$, i.e. the feature space, contrasting with our discussion in Section B. However, this is essentially the only case $\varepsilon_f(u, v)$ simplifies into a feature space distance.

### 3.2 FULLY CONNECTED NEURAL NETWORKS WITH RELU ACTIVATIONS

For this feature extractor architecture, we have $f = f^{(\ell)} \circ \ldots \circ f^{(1)}$ where the layer $i$ function is $f^{(i)} : \mathbb{R}^{n_i} \to \mathbb{R}^{n_{i+1}}$, $f^{(i)}(z) = (k^{(i)} + L^{(i)}z)^+$. Here $z^+$ represents the component-wise positive part of the vector $z$. Then $\varepsilon_f(u, v)$ is the value of the following optimization problem:

$$\min \varepsilon \quad \text{subject to} \quad \delta \in \varepsilon \mathcal{B}, \quad \delta' \in \varepsilon \mathcal{B}, \quad (f^{(\ell)} \circ \ldots \circ f^{(1)})(u + \delta) = (f^{(\ell)} \circ \ldots \circ f^{(1)})(v + \delta').$$

As in the linear case, the minimum exists because the cones are closed and the equality constraint is feasible. In contrast with the linear case, the equality constraint is nonconvex.

The local linear approximation to $f^{(i)}(z)$ around a point $z'$ is $\text{diag}(s^{(i,z')})(k^{(i)} + L^{(i)}z)$, where $s^{(i,z')}$ is a zero-one vector that depends on the sign pattern of $k^{(i)} + L^{(i)}z'$: $s_j^{(i,z')} = \mathbf{1}((k^{(i)} + L^{(i)}z')_j > 0)$. In other words, $s^{(i,z')}$ is the ReLU activation pattern at layer $i$ when $z'$ is the input to that layer.

Using these linear approximations, the feasible set of $(\delta, \delta') \in \mathbb{R}^{n_1 + n_1}$ satisfying the constraint $f(u + \delta) = f(v + \delta')$ can be decomposed as a union of polytopes: each activation pattern defines a linear subspace and there are some linear inequalities specifying the region where that activation pattern actually occurs. For a one-layer network, the linear piece for pattern $s$ is

$$f(x) = \text{diag}(s)(k + Lz) \quad \text{for} \quad \text{diag}(2s - 1)(k + Lz) \geq 0.$$

Thus one of the polytopes composing the feasible set is $(\delta, \delta') \in \mathbb{R}^{n_1 + n_1}$ satisfying

$$\text{diag}(s)(k + L(u + \delta)) = \text{diag}(s')(k + L(v + \delta')),$$
$$(2\,\text{diag}(s) - I)(k + L(u + \delta)) \geq 0,$$
$$(2\,\text{diag}(s') - I)(k + L(v + \delta')) \geq 0.$$

In the one-layer case, the whole feasible region is covered by polytopes where $s = s'$. Observe that the dimension of the subspace satisfying the linear equality varies with $s$. When $f$ contains multiple layers, each polytope in the feasible set is defined by a linear equality constraint involving feature vectors together with a linear inequality for the sign of each ReLU input.

We optimize over this feasible set with a descent algorithm: Algorithm 2 details the version for single-layer networks. The method generalizes to multiple layers and is used to obtain our results in Section 4.2). A full description of the general version is in Appendix C. We initialize our search in a polytope that we know to be nonempty because it contains the feature space collision induced by the midpoint of the two examples. Within each polytope, we can minimize the objective exactly by solving a linear cone program. We examine the dual variables associated with the linear inequality constraints to determine whether an adjacent polytope exists in which we can continue the search.

In the one layer case, the cone program that we solve, which depends on $(L, k, u, v, s)$, is

$$
\begin{aligned}
\min \varepsilon \quad \text{subject to} \quad & (\varepsilon, \delta, \delta') \in \mathbb{R}^{1+n_1+n_1}, \, \delta \in \varepsilon\mathcal{B}, \, \delta' \in \varepsilon\mathcal{B}, \\
& \operatorname{diag}(s)L(u + \delta) = \operatorname{diag}(s)L(v + \delta'), \\
& (2\operatorname{diag}(s) - I)(k + L(u + \delta)) \geq 0, \, (2\operatorname{diag}(s) - I)(k + L(v + \delta')) \geq 0.
\end{aligned}
$$

We need not just the values of the primal solution, but the values of dual variables associated with the linear inequalities in the solution to the dual problem. These are called $z$ and $z'$ in Algorithm 2. When both $z_j > 0$ and $z'_j > 0$, the input to ReLU $j$ is zero when both $f(u+\delta)$ and $f(v+\delta')$ are computed, and the objective function could be decreased further by allowing the input to switch signs. In this case, we move to another polytope based on the new ReLU states and search there.

When we fail to find such pairs of dual variables, either the minimum is in the interior of the current polytope (and thus is supported only by cone constraints), or the minimum is on the boundary of the current polytope but there is no adjacent polytope in which we could continue the search. Thus we end the search.

---

**Algorithm 1** Descent with midpoint initialization

**Input:** $u, v \in \mathbb{R}^{n_1}, L \in R^{n_2 \times n_1}, k \in \mathbb{R}^{n_2}$
**Output:** $\varepsilon \in \mathbb{R}, \delta, \delta' \in \mathbb{R}^{n_1}$
1: $z \leftarrow k + \frac{1}{2}L(u + v), \varepsilon \leftarrow \frac{1}{2}\|u - v\|$
2: **for** $0 \leq j < n_2$ **do**
3: $\quad s_j \leftarrow \mathbf{1}(z_j > 0),$
4: **end for**
5: **repeat**
6: $\quad \varepsilon^{\text{old}} \leftarrow \varepsilon$
7: $\quad (\varepsilon, \delta, \delta', z, z') \leftarrow \text{ConeLP}(L, k, u, v, s)$
8: $\quad s^{\text{old}} \leftarrow s$
9: $\quad$ **for** $0 \leq j < n_2$ **do**
10: $\quad\quad$ **if** $z_j > 0$ and $z'_j > 0$ **then**
11: $\quad\quad\quad s_j \leftarrow 1 - s_j$
12: $\quad\quad$ **end if**
13: $\quad$ **end for**
14: **until** $s^{\text{old}} = s$ or $\varepsilon^{\text{old}} = \varepsilon$

---

**Algorithm termination, convergence, and complexity:** The descent algorithm will terminate in a finite number of iterations and will find a local minimum of the distance function. Since the feasible space is non-convex, any local descent procedure is not guaranteed to find the global minimum. The number of variables in the cone program is proportional to the number of ReLUs in the feature extractor plus the dimension of the example space. The time-complexity of a single iteration of the search is *polynomial in the input dimension and number of ReLUs*. The feasible region is the union of a finite but exponentially large number of convex polytopes. Due to the requirement that each iteration make progress, it is impossible to ever revisit a polytope. Thus, *termination is guaranteed*, but no polynomial bound on the number of iterations is available. We suspect that as in the case of the simplex algorithm for linear programming, input data resulting in an extremely large number of iterations may exist, but would have a delicate structure that is unlikely to arise in practice.

### 3.3 Further common layers

**Convolution:** It is straightforward to represent a convolutional layer as a linear layer by constructing the appropriate Topelitz matrix, and thus our method applies directly.

**Batch-normalization:** Batch norm layers are simply affine functions of their inputs at test-time, and the transformations they induce can be easily included in a linear layer. Our method thus applies.

**Max pooling:** Max-pool layers, like ReLUs, are piecewise linear functions of their inputs, so constraints coming from the equality of max-pool outputs also lead to feasible regions that are the union of polytopes. This is a simple extension left for future work due to a lack of space.

**Other activation functions and architectures:** Injective activation functions such as Leaky ReLU, ELU and sigmoid will not lead to additional collisions. Further, since they are not piecewise, a

different descent algorithm would be needed. We note that our framework cannot find collisions in networks with both forward and backward flows, such as those with attention.

## 4 EVALUATION

In this section, we demonstrate how the methodology outlined in the previous sections for determining the robustness of a fixed feature extractor can be used in practice. Our detailed results provide insights on how the overall robustness of a neural network evolves through its different layers and is impacted by the training procedure used.

### 4.1 FROM COLLISION-FINDING TO LOWER BOUNDS

We find lower bounds on robust loss over the training set by minimizing a loss function over $G_{\mathcal{V}, N, f}$.

**Vertices $\mathcal{V}$ from training data:** The vertex set $\mathcal{V}$ is a representation of the training data. We use 2-class problems derived from MNIST (LeCun & Cortes, 1998) or Fashion MNIST (Xiao et al., 2017) as in previous work (Pydi & Jog, 2020; Bhagoji et al., 2019; 2021).

**Neighborhood function $N$:** We use the common $\ell_2$-norm ball constraint (Madry et al., 2018), in which the adversary's strength is parametrized by the radius $\epsilon$ of the ball [2].

**Fixed feature extractors $f$:** We use the composition of the layers of convolutional and fully-connected DNNs as our fixed feature extractors (details in Appendix E). The first layer of any of these networks (before a non-linear activation is applied) behaves as a linear feature extractor. Subsequent layers behave as non-linear feature extractors. These networks are trained using one of standard cross-entropy loss minimization or robust training with either PGD-based adversarial training (Madry et al., 2018) or the TRADES loss (Zhang et al., 2019).

**Edge set $\mathcal{E}$ from collision finding:** Algorithm 2 provides a greedy, iterative method to find collisions between feature representations of pairs of points from different classes. Each successful collision is an edge in the bipartite conflict graph. Each iteration of this algorithm can be cast as a convex program in the form of a linear cone (as detailed in Appendix D). We solve this linear cone program using CVXOPT (Andersen et al., 2013). Since we are working with an $\ell_2$-norm constrained adversary, we can speed up computation by projecting the inputs onto the space spanned by the right singular vectors of the first linear layer. For linear convolutional layers, we cast the convolution operation as a matrix multiplication to enable the use of the closed form derived in Section 3.1. We also experimented with a modification of the Auto-PGD (Croce & Hein, 2020) algorithm to find collisions at deeper layers with fixed budgets as done in Engstrom et al. (2019). However, this was less effective and more expensive at finding collisions, so all further results use Algorithm 2.

**Computing the lower bound from the conflict graph:** The conflict graph determines the set of possible output probabilities for each vertex (training data point) for the optimal classifier. Minimizing the $0 - 1$ and cross-entropy losses over this graph thus results in a lower bound over these losses since any classifier must incur a larger loss than the optimal classifier, by definition. We use the method from Bhagoji et al. (2021) over the conflict graphs we derive from deeper layer representations. Results in the main body are for the cross-entropy loss (others in Appendix F).

### 4.2 RESULTS

**Interpreting lower bound values:** The lower bound on the input space representations is computed by considering all possible classifiers and is thus the *best possible*, while bounds for other representations are computed by restricting the space of possible classifiers. Intuitively, the latter will always be larger than or equal to the former. The key metric to understanding the robustness of feature extractors is to check *how much larger* the corresponding lower bounds are. The magnitude of this difference measures the contribution of that layer to the overall network's lack of robustness.

**Robustness of linear layer representations:** Using the procedure outlined in Section 3.1, we find collisions after the first linear layer and construct a conflict graph for several models (Fig. 2) for

---

[2]We are aware of the critiques of this constraint (Gilmer et al., 2018) and only use it to compare with previous work.

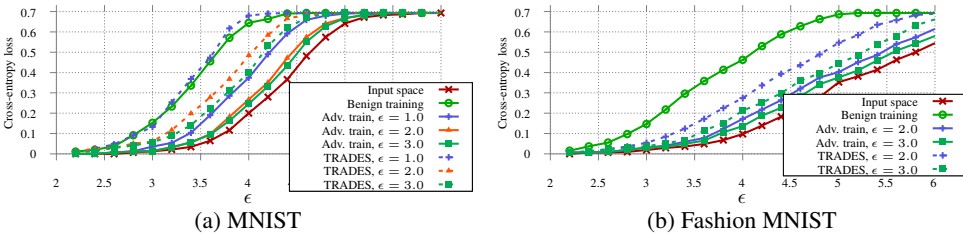

Figure 2: Robustness of the representation obtained from the *first linear layer* of a 3-layer FCNN using different training procedures

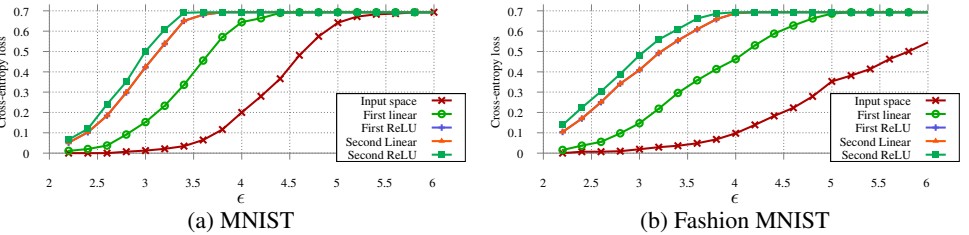

Figure 3: Robustness of representations obtained from different layers of a 3-layer FCNN trained using benign training

both the MNIST and Fashion MNIST datasets. As expected, the linear layers of models with benign training are the least robust, with robustness increasing with the budget $\epsilon_{train}$ used during training. Somewhat surprisingly, the marginal benefits of using a larger budget reduce as $\epsilon_{train}$ increases.

**How does robustness evolve across layers?** Having looked at the how robust linear layer representations are, it is pertinent to consider other representations derived from other layers deeper in the network. Of particular interest are post-ReLU representations for models with benign (Fig. 3) and robust training (Fig. 4). We find that the first ReLU activation layer contributes significantly to an increase in the lower bound for both benign and robustly trained networks. We observe that post-ReLU representations tend to be sparse, leading to an easier search problem and a larger number of collisions. The second linear layer, on the other hand, does not lead to much additional increase in the loss. This is a property of the particular architecture we use, and a smaller linear layer is likely to lead to larger increases in loss. Finally, the second set of ReLU activations does have a measurable impact on robustness, particularly for the benign network. The impact of layer width on robustness is discussed in Appendix F.1.

**How does the parametrization of robust training impact layer-wise robustness?** As expected, layers extracted from robustly trained network are more robust than their benign counterparts for corresponding values of $\epsilon$. We find a significant difference between PGD-based adversarial training and TRADES in terms of the robustness of their first linear layers (Fig. 2), but this largely disappears by the second ReLU activation layer (Fig. 5). Interestingly, we observe a phenomenon where layers of a network robustly trained using higher values of $\epsilon_{train}$ can be *less* robust than those using a lower value, when the value of $\epsilon$ at which evaluation is performed is less than the higher $\epsilon_{train}$.

**Impact of linear convolutional layers on robustness:** As discussed in Section 3.3, the first convolutional layer can be thought as simply a matrix multiplication, although the size of the resulting matrix can be large. We find that since the effective dimension of the resulting features is much larger than the input dimension for the datasets we consider, the linear convolutional layer does not lead to an increase in the lower bound (Fig. 11 in Appendix).

## 5 RELATED WORK AND DISCUSSION

### 5.1 RELATED WORK

**Lower bounds on robustness:** For cases when the data distribution is specified, Dohmatob (2019) and Mahloujifar et al. (2019) use the 'blowup' property to determine bounds on the robust loss, given

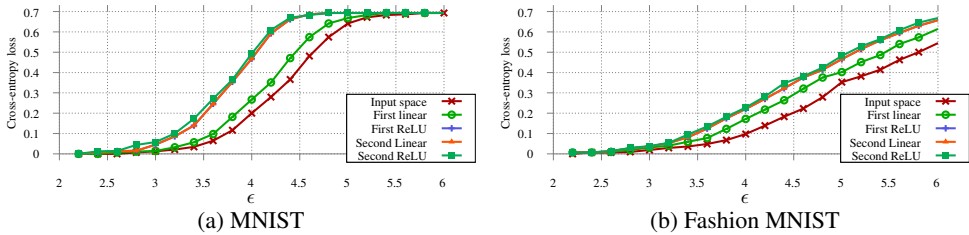

Figure 4: Robustness of representations obtained from different layers of a 3-layer FCNN trained using PGD adversarial training with $\epsilon = 2.0$

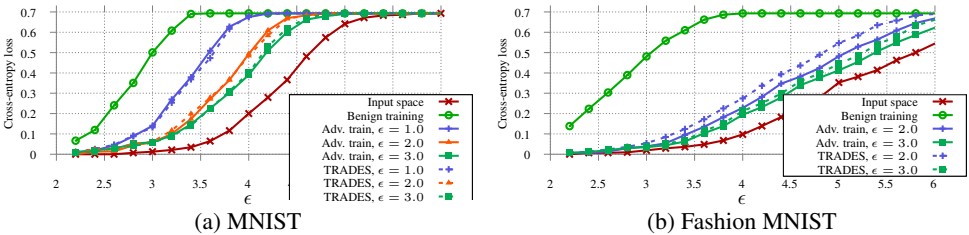

Figure 5: Robustness of the representation obtained from the *second ReLU layer* of a 3-layer FCNN using different training procedures.

some level of loss on benign data. Bhagoji et al. (2019), Pydi & Jog (2020) and Bhagoji et al. (2021) provide lower bounds on robust loss when the set of classifiers under consideration is all measurable functions. These bounds are distribution-agnostic and do not depend on the loss on benign data.

**Certification and Verification:** Work on certified robustness has considered techniques for training neural networks such that the resulting models are provably robust to perturbations upper bounded by a given budget (Kolter & Wong, 2018; Raghunathan et al., 2018; Cohen et al., 2019; Li et al., 2020). Typically, these models can only be certified to be robust to small budgets. In contrast, our work provides lower bounds on the robustness of partial models which are applicable across a large range of budgets. Approaches to verifying the robustness of neural networks (Bunel et al., 2017; Tjeng et al., 2019; Gowal et al., 2018) are closely related to our work, but differ in that they consider fixed end-to-end networks while we focus on a layer-by-layer analysis, allowing us to argue about the robustness of classifiers trained on top of given feature extractors.

## 5.2 DISCUSSION

**Implications for training robust models:** Our results indicate that layers in the network that reduce the effective dimension of their incoming inputs have the largest negative impact on robustness (see further results in Appendix F.1). Two prominent examples of this are ReLU layers that reduce effective dimension by only considering non-negative outputs and fully connected layers with fewer outputs than inputs. On the other hand, linear convolutional layers do not have a negative impact on robustness. This indicates that not reducing the effective dimension of any deeper feature to be lower than that of the input data is likely to benefit robustness. Further, our results confirm that the use of larger values of $\epsilon_{\text{train}}$ does not necessarily translate to higher robustness at lower budgets. This points towards the need for algorithms that can effectively train a network to be robust to more than a single adversary at a time. Finally, we find a qualitative difference in the layers learned using PGD-training and TRADES, implying interesting learning dynamics with different robust losses.

**Extending to state-of-the-art models and datasets:** All of our experiments in this paper are on simple models and datasets which demonstrate the feasibility and use of our method. However, state-of-the art feature extractors for datasets such as Imagenet are far deeper than those considered in this paper. Thus, our algorithm would need to be made considerably faster to handle these cases. While our framework, can handle skip connections, networks with attention are beyond its scope. Nevertheless, the feature extractors we consider *are robust* for the tasks we evaluate them on, making our results and conclusions representative.

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

## A    PROOFS

*Proof of Lemma 1.* Suppose that $e = ((u, 1), (v, -1)) \in \mathcal{E}$. Then, there are some $\tilde{z} \in \mathcal{Z}$, $\tilde{u} \in N(u)$, and $\tilde{v} \in N(v)$ such that $f(\tilde{u}) = f(\tilde{v}) = \tilde{z}$. We have $q_u \leq h(f(\tilde{u}))_1 = h(\tilde{z})_1$, $q_v \leq h(f(\tilde{v}))_{-1} = h(\tilde{z})_{-1}$, and $h(\tilde{z})_1 + h(\tilde{z})_{-1} = 1$. Combining these gives the constraint $(Bq)_e \leq 1$, which appears in (1).

Now, we will show that each vector $q$ in the polytope is achievable by some $h$. The construction is simple: at each point in the feature space, assign the largest possible probability to class 1: let $h(\tilde{z})_1 = \sup_{w:\tilde{z} \in f(N(w))} q_w$ and $h(\tilde{z})_{-1} = 1 - h(\tilde{z})_1$. This achieves the desired performance for examples from class 1:

$$\inf_{\tilde{u} \in N(u)} h(f(\tilde{u}))_1 = \inf_{\tilde{u} \in N(u)} \sup_{w:f(\tilde{u}) \in f(N(w))} q_w \geq \inf_{\tilde{u} \in N(u)} q_u = q_u.$$

For an example is $v$ in class $-1$ we have

$$\inf_{\tilde{v} \in N(v)} h(f(\tilde{v}))_{-1} = \inf_{\tilde{v} \in N(v)} \left( 1 - \sup_{w:f(\tilde{v}) \in f(N(w))} q_w \right)$$
$$= \inf_{\tilde{v} \in N(v)} \inf_{w:f(\tilde{v}) \in f(N(w))} (1 - q_w)$$
$$= \inf_{w:((w,1),(v,-1)) \in \mathcal{E}} (1 - q_w)$$
$$\geq q_v.$$

The final inequality uses the fact that $q$ satisfies $Bq \leq \mathbf{1}$. $\qquad\qquad\qquad\qquad\qquad\qquad\square$

## B    INDUCED DISTANCE IS NOT A DISTANCE ON $\mathcal{Z}$

The fact $\varepsilon_f^*$ is defined on $\mathcal{X}$ is essential. It is not possible to interpret the induced distance as a distance in the feature space $\mathcal{Z}$. This is because $f(N_\varepsilon(x))$, the set of features of points near $x$, cannot be derived from $f(N_0(x))$, the set of features of the uncorrupted version of $x$. This is because distinct choices of $x$ may lead to the same features $f(N_0(x))$, but $f$ may vary more in the neighborhood of one choice of $x$ that the other.

An example is helpful to illustrate this point. Let $\mathcal{X} = \tilde{\mathcal{X}} = \mathbb{R}^2$, let $\mathcal{Z} = \mathbb{R}$, let $N_\varepsilon(x)$ be the closed $\ell_2$ ball of radius $\varepsilon$ around $x$, and let $f(x_0, x_1) = \arctan(x_1/x_0)$ for $x \neq 0$ (the value that we pick for $f(0, x_1)$ is irrelevant). In other words, $f$ finds the angle between the horizontal axis and the line containing $x$. The range of values that the adversary can cause $f(\tilde{x})$ to take depends on $\|x\|$. If $\|x\|_2 < \varepsilon$, $f(\tilde{x})$ can be any angle from 0 to $\pi$, and if $\|x\|_2 \geq \varepsilon$, $|f(\tilde{x}) - f(x)| \leq \arcsin(\varepsilon/\|x\|_2)$.

This example also illustrates that $\varepsilon_f$ is not necessarily a metric, even in cases where $d$ is a metric. Given $u, v \in \mathbb{R}^2$, we have $\varepsilon_f(u, \alpha u) = 0$, $\varepsilon_f(u, \alpha u) = 0$, and $\varepsilon_f(\alpha u, \alpha v)$ can be made arbitrarily small by selecting $\alpha$ to be small. Despite this, $\varepsilon(u, v)$ will be on the order of $\min(\|u\|, \|v\|)$.

An alternative approach to studying the induced distance is to define an adversarial classification problem on $\mathcal{Z}$ by taking $N_\varepsilon^{\mathcal{Z}}(z) = f(N_\varepsilon(f^{-1}(\{z\})))$. Note that this construction only makes sense when $\mathcal{X} = \tilde{\mathcal{X}}$ and thus the domain of $f$ is $\mathcal{X}$. This is more conservative: for any feature point $z$ it considers the worst-case $x$ with feature $z$: the $x$ in whose neighborhood $f$ varies the most.

## C    DESCENT ALGORITHM FOR MULTIPLE LAYER NETWORKS

We start by repeating the notion used in Section 3.2: $f = f^{(\ell)} \circ \ldots \circ f^{(1)}$ where the layer $i$ function is $f^{(i)} : \mathbb{R}^{n_i} \to \mathbb{R}^{n_{i+1}}$, $f^{(i)}(z) = (k^{(i)} + L^{(i)}z)^+$. Then $\varepsilon_f(u, v)$ is the value of the following optimization problem:

$$\min \varepsilon \quad \text{subject to} \quad \delta \in \varepsilon \mathcal{B}, \quad \delta' \in \varepsilon \mathcal{B}, \quad (f^{(\ell)} \circ \ldots \circ f^{(1)})(u + \delta) = (f^{(\ell)} \circ \ldots \circ f^{(1)})(v + \delta').$$

Using the local linear approximation, the equality constraint becomes

$$\text{diag}(s^{(\ell)})(k^{(i)} + L^{(\ell)}(\ldots \text{diag}(s^{(1)})(k^{(1)} + L^{(1)}(u + \delta))))$$
$$= \text{diag}(s'^{(\ell)})(k^{(i)} + L^{(\ell)}(\ldots \text{diag}(s'^{(1)})(k^{(1)} + L^{(1)}(v + \delta')))).$$

and for each $1 \leq i \leq \ell$, the inequalities

$$(2\operatorname{diag}(s^{(i)}) - I)(k^{(i)} + L^{(i)}(\ldots \operatorname{diag}(s^{(1)})(k^{(1)} + L^{(1)}(u + \delta)))) \geq 0$$
$$(2\operatorname{diag}(s'^{(i)}) - I)(k^{(i)} + L^{(i)}(\ldots \operatorname{diag}(s'^{(1)})(k^{(1)} + L^{(1)}(v + \delta')))) \geq 0$$

must hold in order for the linear approximation to be valid. Any collision must be in a polytope with $s^{(\ell)} = s'^{(\ell)}$, which is why we only needed one set of $s$ variables in the single layer case. However, ReLU activation variables for earlier layers cannot be merged.

The main modification to algorithm is to the process of changing the $s$ variables after each iteration. The variables for all but the final layer are allowed to change independently, while the variables in the final layer are changed together following the same rule as in the single layer algorithm.

---

**Algorithm 2** Descent with midpoint initialization

---

**Input:** $u, v \in \mathbb{R}^{n_1}$, $L \in R^{n_2 \times n_1}$, $k \in \mathbb{R}^{n_2}$
**Output:** $\varepsilon \in \mathbb{R}$, $\delta, \delta' \in \mathbb{R}^{n_1}$
1: $z \leftarrow k + \frac{1}{2}L(u + v)$, $\varepsilon \leftarrow \frac{1}{2}\|u - v\|$
2: **for** $0 \leq j < n_2$ **do**
3:     $s_j \leftarrow \mathbf{1}(z_j > 0)$,
4: **end for**
5: **repeat**
6:     $\varepsilon^{\text{old}} \leftarrow \varepsilon$
7:     $(\varepsilon, \delta, \delta', z, z') \leftarrow \text{ConeLP}(L^{(1)} \ldots L^{(\ell)}, k^{(1)} \ldots k^{(\ell)}, u, v, s^{(1)} \ldots s^{(\ell)}, s'^{(1)} \ldots s'^{(\ell-1)})$
8:     $s^{\text{old}} \leftarrow s$
9:     **for** $1 \leq i \leq \ell - 1$ **do**
10:         **for** $0 \leq j < n_{i+1}$ **do**
11:             **if** $z_j^{(i)} > 0$ **then**
12:                 $s_j^{(i)} \leftarrow 1 - s_j^{(i)}$
13:             **end if**
14:             **if** $z_j'^{(i)} > 0$ **then**
15:                 $s_j'^{(i)} \leftarrow 1 - s_j'^{(i)}$
16:             **end if**
17:         **end for**
18:     **end for**
19:     **for** $0 \leq j < n_{\ell+1}$ **do**
20:         **if** $z_j^{(\ell)} > 0$ and $z_j'^{(\ell)} > 0$ **then**
21:             $s_j^{(\ell)} \leftarrow 1 - s_j^{(\ell)}$
22:         **end if**
23:     **end for**
24: **until** $s^{\text{old}} = s$ or $\varepsilon^{\text{old}} = \varepsilon$

---

## D  CONVERTING COLLISION FINDING TO A LINEAR CONE PROGRAM

In this section, we demonstrate how the collision finding problem after one linear and one ReLU layer can be cast as a linear cone program.

We have a network with first layer $v \mapsto Lv + k$ where $L \in \mathbb{R}^{n_1 \times n_0}$ and $k \in \mathbb{R}^{n_1}$.

Given a pair of points $(v', v'') \in \mathbb{R}^{2n_0}$ we would like to search over the space of $(\delta', \delta'') \in \mathbb{R}^{2n_0}$ such that $(L(v' + \delta') - k)^+ = (L(v'' + \delta'') - k)^+$. This space is always nonempty because we can take $v' + \delta' = v'' + \delta'' = (v' + v'')/2$.

Let $S \subseteq [n_1]$ be the subset of active ReLUs. Let $F \in \mathbb{R}^{|S| \times n_1}$ be the inclusion indicator matrix for the subset: $F_{i,j} = 1$ if and only if $j \in S$ and $|S \cap [j]| = i$ (there are exactly $i$ elements of $S$ strictly smaller than $j$, so $j$ is the $i + 1$st element of $S$). Let $D$ be the diagonal matrix with $D_{j,j} = 1$ for $j \in S$ and $D_{j,j} = -1$ for $j \notin S$.

For $j \in S$ (the active ReLUs), we need the constraint $(L(v' + \delta') + k)_j = (L(v' + \delta') + k)_j$. In matrix form, this is $FL(v' + \delta') = FL(v'' + \delta'')$,

For $j \in S$, we need $(L(v' + \delta') + k)_j \geq 0$, $(L(v'' + \delta'') + k)_j \geq 0$, and for $j \notin S$, we need $(L(v' + \delta') + k)_j \leq 0$, $(L(v'' + \delta'') + k)_j \leq 0$. In matrix form, these are $D(L(v' + \delta') + k) \geq 0$, $D(L(v'' + \delta'') + k) \geq 0$.

Our objective is $\max(\|\delta'\|_2, \|\delta''\|_2)$. We will replace with a linear objective by adding two additional variables $(t', t'')$ satisfying $t' \geq \|\delta'\|_2$ and $t'' \geq \|\delta''\|_2$.

The cvx-opt system is

$$\text{minimize } c^T x$$
$$\text{subject to } Gx + s = h$$
$$Ax = b$$
$$s \succeq 0$$

The cones involved are a nonnegative orthant of dimension $2n_1$ and two second order cones, each of dimension $n_0 + 1$. Thus $s = (D(L(v' + \delta') + k), D(L(v'' + \delta'') + k), t', \delta', t'', \delta'')$. We then let $x = (t, \delta', \delta'')$. The matrix relating $s$ and $x$ is $G \in \mathbb{R}^{(n_1 + n_1 + 1 + n_0 + 1 + n_0) \times (1 + n_0 + n_0)}$ and $s = h - Gx$. The block structure is

$$G = \begin{pmatrix} 0 & -DL & 0 \\ 0 & 0 & -DL \\ -1 & 0 & 0 \\ 0 & -I & 0 \\ -1 & 0 & 0 \\ 0 & 0 & -I \end{pmatrix} \qquad h = \begin{pmatrix} D(Lv' + k) \\ D(Lv'' + k) \\ 0 \\ 0 \\ 0 \\ 0 \end{pmatrix}$$

We use $Ax = b$ to encode $FL(\delta' - \delta'') = -FL(v' - v'')$, so $A \in \mathbb{R}^{|S| \times (1 + n_0 + n_0)}$, $b \in \mathbb{R}^{|S|}$, and

$$A = (0 \quad FL \quad -FL) \qquad b = -FL(v' - v'').$$

Finally,

$$c = \begin{pmatrix} 1 \\ 0 \\ 0 \end{pmatrix}.$$

### D.1 Factorized $L$

If $n_1 < n_0$, then we can take advantage of the fact that the rank of $L$ is at most $n_1$. Let $L = U\Sigma V^T$.

Replace $L(v + \delta)$ with $U\Sigma(V^T v + \epsilon)$, $n_0$ with size of $\Sigma$.

New $G$ and $A$:

$$G = \begin{pmatrix} 0 & -DU\Sigma & 0 \\ 0 & 0 & -DU\Sigma \\ -1 & 0 & 0 \\ 0 & -I & 0 \\ -1 & 0 & 0 \\ 0 & 0 & -I \end{pmatrix} \qquad A = (0 \quad FU\Sigma \quad -FU\Sigma)$$

Also, the length of $c$ is reduced. The constant vectors $h$ and $b$ are unchanged.

## E Further experimental details

We consider the following two architectures for our experiments:

1. 3-layer fully connected neural network (FCNN): 300 FC-ReLU-200 FC-ReLU-2 FC
2. 4-layer convolutional neural network (CNN): 20×5×5 conv.-BN-ReLU- 2×2 MaxPool- 50×5×5 conv.-BN-ReLU- 2×2 MaxPool- 500 FC - 2 FC

We also construct a 'Small' and 'Smaller' version of the FCNN with layers that are $2\times$ and $4\times$ narrower respectively.

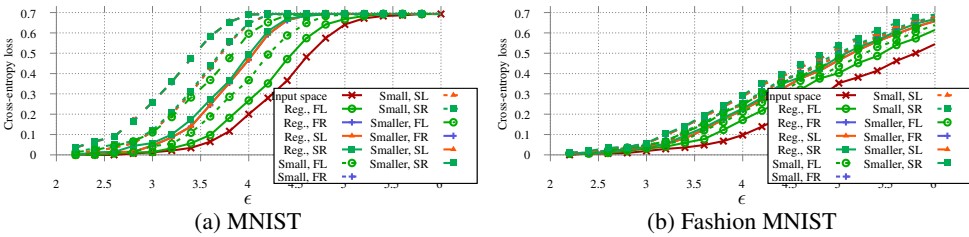

Figure 6: Robustness of the representations obtained from fully connected networks with layers of decreasing size.

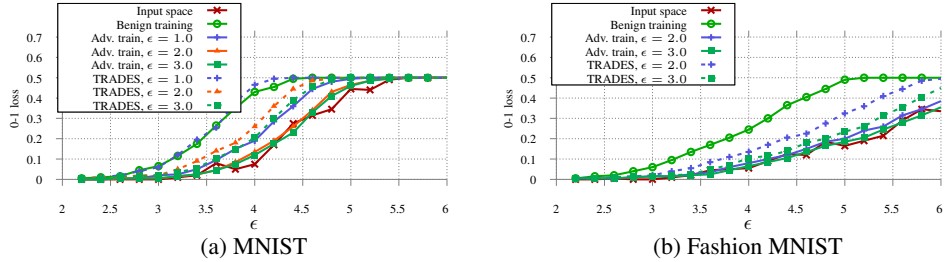

Figure 7: $0 - 1$ Loss: Robustness of the representation obtained from the *first linear layer* of a 3-layer FCNN using different training procedures

# F    ADDITIONAL RESULTS

## F.1    IMPACT OF REDUCTION OF NETWORK SIZE

In Figure 6, we compare the robustness of representations obtained from fully connected networks with decreasing layer sizes. The 'Regular' network is the one used throughout, while the 'Small' and 'Smaller' networks have corresponding layers that are $2\times$ and $4\times$ narrower respectively. We can clearly see that as the width of the feature extractor decreases, so does its robustness.

## F.2    $0 - 1$ LOSS RESULTS

In Figures 7, 8, 9 and 10 we provide lower bounds on the $0 - 1$ loss in the same settings as those considered in the main body of the paper. We note that the results and conclusions remain the same qualitatively.

## F.3    LINEAR CONVOLUTIONAL LAYERS

In Figure 11, we can see that the representations extracted from the first linear layer of a convolutional network do not have any negative impact on the robustness of the overall model.

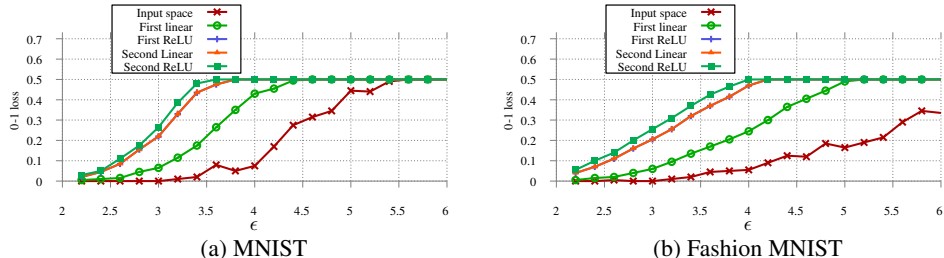

Figure 8: $0 - 1$ Loss: Robustness of representations obtained from different layers of a 3-layer FCNN trained using benign training

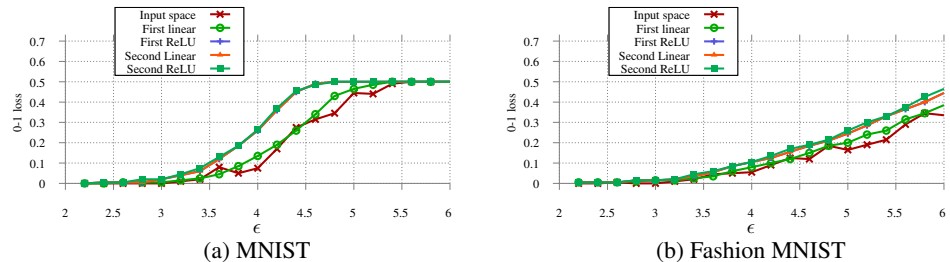

Figure 9: $0 - 1$ Loss: Robustness of representations obtained from different layers of a 3-layer FCNN trained using PGD adversarial training with $\epsilon = 2.0$

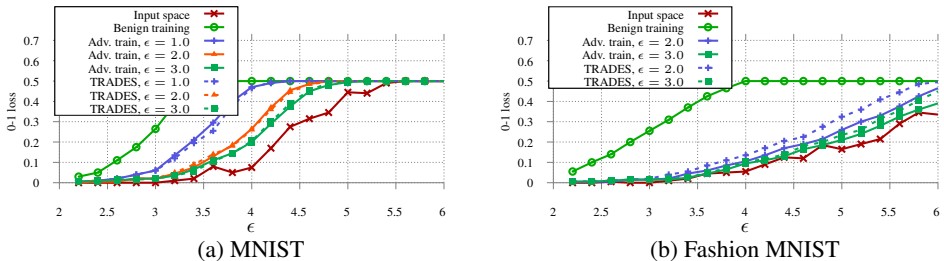

Figure 10: $0 - 1$ Loss: Robustness of the representation obtained from the *second ReLU layer* of a 3-layer FCNN using different training procedures.

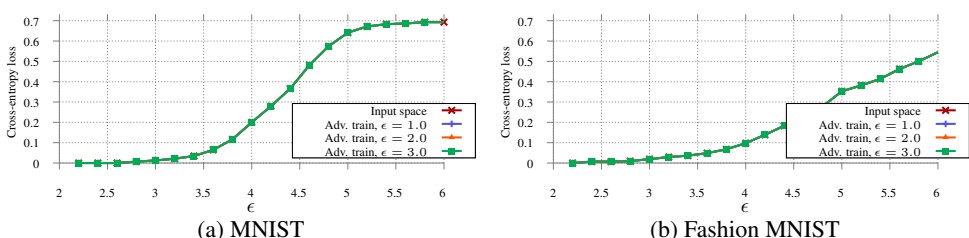

Figure 11: Robustness of the representation obtained from the *first linear layer* of a 4-layer CNN using different training procedures.

