# OpenReview forum: "Lower Bounds on the Robustness of Fixed Feature Extractors to Test-time Adversaries"
_ICLR.cc/2022/Conference — ICLR 2022 Submitted_

### Official Review · Reviewer_hh5w · 2021-11-02

**Correctness:** 3
**Technical Novelty And Significance:** 3
**Empirical Novelty And Significance:** 3
**Recommendation:** 6
**Confidence:** 3

**Main Review:**

Overall I like the idea of the paper: it gives more insights to how different model architecture can affect the robustness lower bound, and thus gives some guidance to how we design a robust system.

On the other hand, I have difficulty understanding the theoretical part and the algorithmic aspect of the paper. The paper is written in a rather theoretical manner. I hope the authors can comment on my following questions so that I can understand the details better.

1) Does the robustness lower bound depend on the underlying data distribution? Intuitively, the collision graph shows that whether two instances in the space can have the same representation after perturbation. However, it's possible that 1) u, v have collision but 2) u, v will never appear in natural samples. I don't see an explicit link to the input distribution, and therefore hope the authors to clarify what this robust entropy is measuring.

2) Are the robust entropy evaluated on the training dataset? What's the relation of this empirical value to the theoretical value over the distribution? I know that the full graph will only have more collision, but the probability of each node appearing can change too. Therefore, the relation is not immediately clear to me.

3) In Section 3.1, the algorithm solves an optimization problem using descent. My questions is, how does the outcome of this algorithm translate to the figures in Sec 4? Or rather, how is Figure 2 produced out of the outcome of Algorithm 1? (Sec 4 gives a pointer to Bhagoji et.al, but it would be good to summarize the key steps in this procedure.)



**Summary Of The Paper:**

This paper investigates the lower bound of robustness, i.e. the best robustness we can expect, with the added constraints of model architecture. It derives an algorithm of estimating the lower bound, and empirically shows how various types of layers -- such as linear, ReLU and convolutional layer -- can impact the lowerbound.

**Summary Of The Review:**

Overall I find the idea interesting and the results reasonable. Once my questions are addressed, I can offer better advice on the paper organization. I'm giving a score of 5 now, but I'm willing to raise the score after my questions are answered.

===================================================

After viewing the author response, my technical questions are resolved. I'm raising the score to 6 for the completeness in the idea. The paper can be enhanced by considering considering more data sets. MNIST and FashionMNIST is slightly too simple. The in-sample v.s. distributional robustness can be an interesting future direction to explore too. Currently it's addressed by using a practical attack algorithm as a probe. More explanation to this end can enhance the paper too.

---

### Official Review · Reviewer_8XE9 · 2021-11-02

**Correctness:** 4
**Technical Novelty And Significance:** 3
**Empirical Novelty And Significance:** 3
**Recommendation:** 8
**Confidence:** 3

**Main Review:**

Strengths:

- This paper expands on previous theory while making it directly applicable to neural networks that are commonly used.
- They provide an algorithm that calculates lower bounds on robustness for real networks.
- Experimental results on multiple datasets and robust training algorithms are provided, comparing how close the different methods are to their theoretical lower bounds.
- Based on their experimental results, they discuss implications for model training. Specifically they note that ReLU seems to reduce the robustness of features in later layers of the network.

Weaknesses:

- There is no discussion of other network architectures for which this method does not work. The method seems to work on the fact that ReLU layers reduce the dimensionality of the space that must be searched for collisions, so it would be interesting to hear about other activation functions for which that is not the case. The subsection "Other neural network architectures" only seems to cover networks where this method works. It also seems to require that the entire network (including activation functions) be expressible as a series of matrix multiplications, so I'm not sure of the applicability of this to architectures with, for example, attention.

- Some notation and definitions are unclear or hard to follow. For example, in section 3: "We take the neighborhood of any point to be a scaled shifted version of this ball: N_\epsilon(x): n + \epsilon \mathcal{B}", but x is missing from the right hand side.

In subsection 3.2 "... satisfying the constraint f(u + \delta) = f(v + \delta)" seems like it should be "f(u + \delta) = f(v + \delta')". \epsilon seems to be a bit overloaded as well, where it is a function of two points in section in 2.1 and later a real number.

In equation 1 in lemma 1, since q is a vector, are the inequalities acting coordinate-wise on the vector? In that case, is q a vector in \[0, 1\]^\mathcal{V}? I have a similar question regarding the equation in definition 1, as q is also a vector in an inequality with a real number and another vector.


**Summary Of The Paper:**

This paper adapts previous work on the lower bounds  of robustness from any possible measurable function to commonly used neural architectures for image classification. In order to do this, they create an algorithm based on linear programming to find collisions in feature representations at different layers in neural networks. Once the collisions are found, the minimum achievable loss by a classifier using these features is computed. Experiments are done on fully connected neural networks trained on MNIST and Fashion MNIST with standard and robust training methods. The authors find that adversarial training produces features closer to their predicted lower bound than networks trained with TRADES loss or with standard training.

**Summary Of The Review:**

This paper provides an important application of previous work while also introducing a practical algorithm to find collisions in feature space and calculate a lower bound of adversarial robustness. While neural network architectures for which this algorithm is not applicable seem to be ignored and some notational issues made it take longer to read, I think the paper should be accepted.

---

### Official Review · Reviewer_nQgV · 2021-11-03

**Correctness:** 3
**Technical Novelty And Significance:** 3
**Empirical Novelty And Significance:** 2
**Recommendation:** 6
**Confidence:** 3

**Main Review:**

Strength:
- The authors develop a reasonable algorithm for constructing the conflict graph for a specific fixed feature extractor.

Weakness:
- Motivation of calculating the lower bound on the adversarial loss for the fixed feature extractor is somewhat unclear.
- The paper lacks the analysis for justifying the proposed algorithm.


My current decision is acceptance. However, the motivation of the fixed feature extractor is unclear for me. I'd like the authors to clarify the motivation for calculating the lower bound on the adversarial loss with the fixed feature extractor.

The motivations in the related works, including Pydi & Jog 2020 and Bhagoji et al. 2019;  2021, are understandable. The lower bounds on the Bayes optimal adversarial loss are useful for evaluating the performance of the robust learning algorithms and for confirming the fundamental limit of the robustness. The lower bound investigated in this paper is no longer the fundamental limit. Because there is no necessity to employ the pre-trained feature extractor, I cannot imagine the situation where the use of the given feature extractor is enforced. So, I'm wondering if such a lower bound is useful for evaluating something. In the experiments, the authors evaluated the lower bounds with fixed first 1 or 2 layers as usage of the proposed method; however, it is unclear how to use the evaluated adversarial loss.

The theoretical result, i.e., Lemma 1, is the straightforward extension of Bhagoji et al. 2021's result. The theoretical contribution of this paper seems to be incremental.

The main contribution of this paper is the algorithm development shown in Section 3. The authors provide the reasonable construction of the algorithm for certifying the existence of the specific edge in the conflict graph. However, it is non-trivial that the present algorithm converges to the desired solution. It is necessary to clarify this point.


Minor comments:
- In the second paragraph on page 3, the right bracket is missing in the adversarial loss.

**Summary Of The Paper:**

This paper investigates the lower bound on the adversarial loss with the given specific feature extractor.  The authors develop an algorithm to compute such a lower bound for the feature extractor employing the forward feed network with ReLU activation. The experiments demonstrate the usage of the proposed algorithm.

**Summary Of The Review:**

My current decision is acceptance. However, the motivation of the fixed feature extractor is unclear for me. I'd like the authors to clarify the motivation for calculating the lower bound on the adversarial loss with the fixed feature extractor.

---

### Official Review · Reviewer_r9vz · 2021-11-03

**Correctness:** 3
**Technical Novelty And Significance:** 2
**Empirical Novelty And Significance:** 2
**Recommendation:** 6
**Confidence:** 3

**Main Review:**

Please find my full review of the paper *"Lower Bounds on the Robustness of Fixed Feature Extractors to Test-Time Adversaries"* below. Unfortunately I can not recommend the current version of the presented manuscript for acceptance at the ICLR conference. The main contributors to this decision is the unclarity in the contributions of the work, the empirical evidence supporting the claims made being too sparse, and the large amount of crucial work left for future work.

Before further elaborating on my reservations about the paper I would like to clearly state that I found the paper to be particularly hard to follow. Personally I believe that this is primarily due to the introduction not properly introducing the setting and aims of the paper, but I should also acknowledge that I was not entirely familiar with the work [Bhagoji, 2021] (on which this paper heavily builds) before getting assigned this paper for review.
For this reason, I would be very interested in hearing from the other reviewers to what extend they found the paper hard to follow as well.

### Contributions of work

If I understood correctly, the main contribution of the paper is to provide a lower bound on the adversarial robustness of specific models. This is in contrast with earlier work on adversarial robustness certification that focusses on arbitrary classification functions. While this does sound interesting, I struggle in finding how this differs from performing a simple adversarial robustness evaluation on a model using a standard adversarial attack. This should also provide a lower-bound on the true loss of the model under attack. I would be grateful if the authors could clarify this for me.

The method by which the authors achieve this lower-bound heavily builds upon earlier work [Bhagoji, 2021]. Again, if I understood correctly, the main contribution of the work presented here is extending this method to small (mostly linear) neural networks using a greedy search algorithm. Given that there is not true evaluation presented of how efficient this search algorithm is, I find it difficult to evaluation the significance of this contribution.

### Empirical Evidence

The authors use their proposed method for determining a number general statements about layers that hypothetically lead to a measurable drop in robustness. Based on the rigour of the presented empirical evaluation, I do not believe that these general claims can be made. In general, the method proposed by the authors is only applied to a very limited set of network architectures. The models evaluated in the paper are in general very shallow and in my opinion not representative of architectures used in practice. For example, the architecture used is only applicable for binary classification.

The fact that this limitation of the architecture is not clearly communicated in the main body of the paper is something that I find to be especially troubling.

### Future work

Lastly, I believe that the current version of the manuscript, unfortunately, leaves most of the interesting for future work. For example, the extension to CNNs, BatchNormalization and Residual Blocks is left for future work. With these often applied layers pushed to future work, I find it hard to judge the applicability of the presented method.


# Updates after rebuttal
I've increased my score. Please see the comment chain for further clarification.

**Summary Of The Paper:**

The authors present a method for deriving a lower bound on the loss of a specific model architecture under adversarial attack. The method heavily relies on previous work [Bhagoji, 2021]. The main contribution of the paper is therefore extending the concepts presented in [Bhagoji, 2021] for evaluating the input of a model to the output of later layers.
Using their proposed method the authors perform an evaluation of the robustness of a simple feature extractor.

**Summary Of The Review:**

The contributions of the paper are unfortunately unclear. In addition to this, based on the empirical evaluation, the made claims are to general.

---

### Official Review · Reviewer_btdg · 2021-11-04

**Correctness:** 3
**Technical Novelty And Significance:** 3
**Empirical Novelty And Significance:** 3
**Recommendation:** 5
**Confidence:** 4

**Main Review:**

The paper begins by analyzing linear feature extractors, which is a simpler case that can be analyzed theoretically in full, but is not directly applicable to the non-linear case. For the non-linear case, the paper uses gradient descent to try to find collisions, which is reasonable; the authors may want to include a citation to [1], which has also previously tried to find collisions in representations via a different gradient descent algorithm on inputs.

The main algorithm (Algorithm 1 in the paper) appears reasonable, but I have some questions about it. In particular, in line 10, it appears to cause branching for every ReLU. How does this algorithm scale with the number of ReLUs? Will it scale exponentially? If so, it is worth mentioning, as this is often a limitation of related methods that involving branching for every ReLU. For example, exact certification of neural network robustness is shown to be NP-hard in [2].

The experimental results in this work are primarily focused on very basic datasets (MNIST and Fashion-MNIST), so it is unclear how results will generalize to larger datasets (CIFAR, ImageNet). The networks used are also usually simple 3 layer networks; I think the paper could be made more convincing with either (1) experiments on more complex datasets/models and/or (2) discussion/experiments showing how this approach scales as model size/dataset complexity increases. For example, how does the proposed algorithm scale for deeper networks? Could it feasibly be scaled up to find collisions for ResNets or deeper ConvNets?

Finally, the authors draw conclusions regarding how various layers contribute to (lack-of) robustness. I think this section is very interesting, but I think deeper investigation is necessary to draw such conclusions. In particular, the authors focus on how their robustness loss lower bound changes depending on the structure of each layer, but this robustness lower bound is merely a proxy for true robustness. Thus, concluding that certain properties of each layer affect true robustness is too strong, and should be qualified further. In particular, the implications for how to train robust models section is interesting, and it may be worthwhile to try to find supportive experimental evidence to see whether insights derived from analyzing the robustness lower bound help.

Small Comments and Questions:
- You repeat “are fixed” before and after the parentheses in the 4th-to-last line of page 1
- In Figure 3a/b, the blue line is missing
- You mention that “convolutional networks are more robust than fully connected ones” is “commonly held wisdom” - is this actually true? Usually, adversarial attacks are capable of completely breaking both types of networks.

[1] https://arxiv.org/abs/1906.00945 Adversarial Robustness as a Prior for Learned Representations (Logan Engstrom, Andrew Ilyas, Shibani Santurkar, Dimitris Tsipras, Brandon Tran, Aleksander Madry)

[2] https://arxiv.org/abs/1702.01135 Reluplex: An Efficient SMT Solver for Verifying Deep Neural Networks
Guy Katz, Clark Barrett, David Dill, Kyle Julian, Mykel Kochenderfer


=========================================================================

I thank the authors for the rebuttal. After reading the rebuttal and other reviewer comments, I have decided to maintain my score of "weak reject". As Reviewer r9vz noted, a lot of the most interesting work is left as "future work" at the moment; it is still unclear to me whether this approach will scale effectively to more complex datasets (CIFAR/ImageNet vs. MNIST/Fashion-MNIST) or more standard and modern architectures (CNNs, resnets vs. 3-layer fully-connected networks).

**Summary Of The Paper:**

This paper analyzes the robustness of fixed feature extractors (e.g. all but the last layer of a pre-trained network). They do so by attempting to find pairs of inputs that have collisions (at some later layer of the network). By analyzing these collisions in both simpler cases and in more general cases, the authors argue that they can also pinpoint what properties of certain layers contribute to lack-of-robustness, including dimensionality reducing layers, different adversaries at train/test, and ReLUs.


**Summary Of The Review:**

I recommend a weak reject due to the fact that the datasets and models used in this paper are too simplistic; understanding how this method scales as those factors change will improve the work.

---

### Decision · Program_Chairs · 2022-01-20

**Decision:**

Reject

**Comment:**

Thank you for your submission to ICLR.  The reviewers were split on this paper, with more favoring acceptance but with relatively low confidence.  After reading through the paper and reviews, I tend to agree slightly more with the more critical comments.  The paper is very much on the borderline, but ultimately 1) the rather incremental nature of the work compared to [Bhagoji, 2021], and 2) the rather small-scale evaluations in the current version, which the field has largely moved on from as they often give overly-optimistic impressions of certified robustness.  Ultimately a lot of the extensions (which at this point are fairly standard in most methods for deep network verification), seem like they should really be taken into account in the current paper.  For these reasons I lean slightly towards not accepting the paper in its current state.